# Wind Turbine Gearbox Gear Surface Defect Detection Based on Multiscale Feature Reconstruction

**Rui Gao** [1,*]**, Jingfei Cao** [1,*]**, Xiangang Cao** [2]**, Jingyi Du** [1]**, Hang Xue** [1] **and Daming Liang** [1]

[1] College of Electrical and Control Engineering, Xi'an University of Science and Technology,
Xi'an 710054, China; 000248@xust.edu.cn (J.D.)

[2] School of Mechanical Engineering, Xi'an University of Science and Technology, Xi'an 710054, China;
caoxg@xust.edu.cn

* Correspondence: gaorui@xust.edu.cn (R.G.); 20206043045@stu.xust.edu.cn (J.C.)

**Abstract:** The fast and accurate detection of wind turbine gearbox surface defects is crucial for wind turbine maintenance and power security. However, owing to the uneven distribution of gear surface defects and the interference of complex backgrounds, there are limitations to gear-surface defect detection; therefore, this paper proposes a multiscale feature reconstruction-based detection method for wind turbine gearbox surface defects. First, the Swin Transformer was used as a backbone network based on the PSPNet network to obtain global and local features through multiscale feature reconstruction. Second, a Feature Similarity Module was used to filter important feature sub-blocks, which increased the inter-class differences and reduced the intra-class differences to enhance the discriminative ability of the model for similar features. Finally, the fusion of contextual information using the pyramid pooling module enhanced the extraction of gear surface defect features at different scales. The experimental results indicated that the improved algorithm outperformed the original PSPNet algorithm by 1.21% and 3.88% for the mean intersection over union and mean pixel accuracy, respectively, and significantly outperformed semantic segmentation networks such as U-Net and DeepLabv3+.

**Keywords:** wind turbine gearbox; defect detection; multiscale feature reconstruction; feature selection





## 1. Introduction

### 1.1. Wind Turbine Gearbox Defect Detection Methods

Wind turbines are often located in fields, and the internal components of the gearbox are prone to defects due to contact fatigue or the entry of external objects. When a serious failure occurs, it will directly affect the production operations or even the equipment downtime, significantly impacting the power supply and business efficiency [1]. Therefore, the internal structure of the gearbox must be inspected regularly—particularly, transmission structures such as gears and bearings. The inspection methods can be divided into two types, i.e., contact and noncontact, depending on whether detection devices need to be installed. The contact inspection methods require the installation of additional sensors or data acquisition devices to monitor the gearbox operation data in data-monitoring situations, whereas the noncontact inspection methods mainly use nondestructive equipment, such as industrial endoscopes, to visually inspect the interior of gearboxes and monitor defects of gears and bearings.

Typically, in contact detection methods, the vibration signals of a gearbox gears and bearings are analyzed, and the locations of defects are determined using the mesh frequency of the gears. Wu et al. [2] proposed a composite multiscale entropy (CMSE) method for extracting features from faulty gear vibration signals and experimentally verified that the feature extraction capability of the method was superior to that of the multiscale entropy method. Zhao et al. [3] used two new nonlinear dynamic methods, i.e., local maximum scale entropy and extended multiscale entropy, which could effectively extract fault features

from rolling–bearing vibration signals. Gunasegaran et al. [4] investigated the problem of local fault detection in helical gear systems, proposed a vibration and acoustic signal evaluation method based on higher-order spectral analysis, and experimentally verified its effectiveness and reliability. Although the contact detection method uses vibration signals and can reliably diagnose most defects, the vibration signals are sensitive to external noise, which can easily cause "Related Pseudo Signals ", that is, the vibration signals of multiple gears or bearings and the background noise are interspersed with the vibration signals of a single measurement point. Therefore, the vibration signals are complex and cannot be used to directly detect defects on the gear surface [5]. ZamudioRamirez Israel et al. [6] presented a non-invasive, gradual wear diagnosis method for bearings outer-race faults. This technique relies on the application of a linear discriminant analysis (LDA) to statistical and Katz's fractal dimension features obtained from stray flux signals, and then an automatic classification is performed by means of a feed-forward neural network (FFNN). Israel Zamudio-Ramirez et al. [7] proposed a methodology based on the processing of stray flux signals through feature calculation and extraction stages that lead to a high-performance signal characterization by estimating a set of statistical time domain-based features, then reduced dimensionally by means of principal component analysis and linear discriminant analysis techniques.

In noncontact inspection methods, the inability to detect defects directly is overcome by photographing the inside of wind turbine gearboxes using industrial endoscopes, which are often used in conjunction with contact inspection methods to detect defects. Zou et al. [8] used industrial endoscopes to inspect gearbox defects and explore the primary causes of failure during gearbox operations. Sun et al. [9] proposed AIFN-IA, which is based on the complementary fusion of image and acoustic data, and used it to identify six types of structural and nonstructural damages in gearboxes, achieving a high detection accuracy. Zhou et al. [10] combined vibration analysis and image acquisition equipment to develop a gearbox fault diagnosis method using the fusion of vibration signals and infrared images. The proposed method not only maintains high robustness but also can identify structural and nonstructural health states. In the aforementioned studies, manual analysis or traditional image processing methods were mainly used to detect defects, which are more intuitive than other methods; however, the manual adjustment of features in image processing suffers from the problem of incomplete feature extraction and is easily disturbed by environmental factors in different application scenarios [11]. The studies on detection methods are summarized in Table 1.

**Table 1.** Works on gearbox fault detection methods.

|  | Authors | Methods |
| --- | --- | --- |
| contact inspection | Shuen De Wu [2] | Vibration Signal |
|  | Dongfang Zhao [3] | Vibration Signal |
|  | Gunasegaran V [4] | Vibration Signal |
|  | ZamudioRamirez Israe [6] | Stray flux signals |
|  | Israel Zamudio-Ramirez [7] | Stray flux signals |
| noncontact inspection | Zou Xiangfu [8] | Images |
|  | Sun Dingyi [9] | Images |
|  | Zhou Qiting [10] | Images |

*1.2. Deep Learning-Based Image for Defect Detection*

In recent years, deep learning has allowed significant progress in image classification, target detection, and semantic segmentation. Owing to its end-to-end advantages, deep learning effectively reduces manual interventions in training and maintains a stable detection accuracy for defect detection. With regard to defect classification, Xu et al. [12] proposed a deep-learning network based on a point cloud representation for detecting gear defects, which can accurately extract local information on gears. Su et al. [13] proposed a small-sample gearbox defect diagnosis method based on an improved generative adversar-

ial network. A weak learner was established to maximize the deep-learning efficiency, and the K-nearest neighbor algorithm and two-stream convolutional networks were used to score and fuse fault data to achieve fault classification and diagnosis. The method had a high diagnostic accuracy and classification accuracy in small-sample set fault diagnosis of wind turbine gearboxes. Nath et al. [14] introduced the NSLNet framework using ImageNet as a feature extraction module and performed adversarial training through neural networks in the extracted feature space to accurately detect defects.

Regarding research on the localization of defects, Abdelrahman et al. [15] used an R-CNN network for the detection of gear defects, which reduced the manual detection by 66% and achieved 88% accuracy with 86% recall. Yu et al. [16] proposed a gear defect online detection model called S-YOLO and found that it allowed a better recognition of microdefects in complex backgrounds than the YOLOv3 target recognition network. Qin et al. [17] proposed a three-stage knob gear recognition method based on YOLOv4 and the Darknet53-DUC-DSNT model and for the first time applied deep-learning key point detection to knob gear recognition, effectively improving the gear. Xie et al. [18] designed a lightweight KD-EG-Rep VGG network based on structural reparameterization to detect surface defects on strip steel; the model had a defect recognition accuracy of 99.44% on the test set and a detection time of 2.4 ms for a single image, with a high accuracy and good generalization capability. Wang et al. [19] proposed an improved MS-YOLOv5 model based on the YOLOv5 algorithm for problems involving low recognition rate, random defect distribution, and large variation for the detection of surface defects in aluminum profiles, which achieved defect detection with a high average accuracy.

Regarding research on segmentation for defect detection, semantic segmentation refers to the implementation of pixel-level classification in an image to achieve the accurate identification and localization of different targets in the image. Semantic segmentation is more complex and accurate for pixel-level classification than for image classification, where categories are assigned to the whole image, and targets are detected by framing them. Xiao et al. [20] used the SHGA-PSO algorithm to detect metallurgical gear defects. Compared with the GA-BP, PSO-BP, and GA-PSO-BP algorithms, the SHGA-PSO-BP algorithm for defect diagnosis exhibited not only a stronger generalization ability but also a higher recognition accuracy. However, in practical defect segmentation, because the operation effect of a traditional CNN is limited by the size of the convolution kernel, the network can only obtain limited local information and cannot effectively utilize global information. Zhao et al. [21] proposed the Pyramid Scene Analysis Network (PSPNet), which makes full use of the contextual information of different regions through a pyramid pooling module (PPM), thus solving the spatial information loss problem and achieving good results in segmentation tasks. Yuan et al. [22] used the PSPNet network to extract buildings from remote-sensing images and panned the pooled PSPNet so that the pixels located at the edges of the network could obtain the features of the entire local area with a high segmentation accuracy. Wang et al. [23] achieved the fast detection of incomplete coal and gangue by improving the PSPNet via the introduction of an attention mechanism, atrous convolution, and depthwise separable convolution. The average cross-merge ratio of the final segmentation was 95.4%, which was better than those of other models. The studies on deep learning-based image defect detection are reported in Table 2.

In summary, there has been more research on defect detection using deep learning methods, and better results have been achieved. However, the following problems still exist in wind turbine gearbox surface defect detection:

(1) The feature extraction downsampling operation in the CNN-based detection method reduces the feature image resolution and causes a loss of information regarding cracked gear defects.

(2) The presence of a large amount of repetitive and cumbersome feature information in the gear surface dataset interferes with the extraction of image features, requiring the accurate screening of important features and the removal of redundant features.

(3) Factors such as image dithering and complex image backgrounds in image acquisition affect the detection accuracy.

**Table 2.** Works on deep learning-based image defect detection.

| | Authors | Algorithms | Disadvantages |
|---|---|---|---|
| classification | Xu Zhenxing [12] | Point-cloud representation | Applicable to classification, no positioning |
| | Su Yuanhao [13] | Improved generative adversarial network | Large models |
| | Nath Vikanksh [14] | NSLNet | Unable to mark defect location |
| object detection | Allam Abdelrahman [15] | R-CNN | Needs exztensive image training |
| | Yu Liya [16] | S-YOLO | Large difference between background and target is required |
| | Qin Ronglin [17] Xie Xu [18] Wang Teng [19] | YOLOv4, Darknet53-DUC-DSNT KD-EG-RepVGG MS-YOLOv5 | Large models, slow detection speed Bias in localization Large model, slow detection speed |
| Segmentation | Xiao Maohua [20] Hengshuang Zhao [21] Yuan Wei [22] | SHGA-PSO PSPNet Improved PSP Network | Complex steps, difficult to operate Unable to extract the edge features of an image False detection in complex backgrounds |
| | Wang Xi [23] | Improved PSP Network | Misjudgment when the background is similar to the target |

## 2. Multiscale Feature Reconstruction for a Gearbox Surface Defect Detection Model

To address the problems of limited feature extraction capability and similarity of defect feature information in gear surface defect detection, this paper proposes a multiscale feature reconstruction defect segmentation method based on the PSPNet model with Swin Transformer as the backbone network. The overall framework of the model is shown in Figure 1. The main steps of the model are as follows:

(1) In the multiscale feature reconstruction stage, the Swin Transformer (a hierarchical vision transformer using shifted windows) [24] network model was used as the multiscale feature reconstruction structure, and the multiscale feature extraction problem was solved using shifted-window self-attention and the block merging layer. The global feature map was obtained by stitching multiple local feature maps together, and then the multiscale features were extracted and reconstructed.

(2) In the feature selection stage, a Feature Similarity Module (FSM) was used to select the most useful features from the original features and enhance the ability of the model to extract nuanced features.

(3) In the multiscale feature fusion stage, the PPM was used to fuse the output of the PPM with the features extracted by the feature extraction module and obtain a feature map that incorporated multiscale information.

(4) In the output stage, a fully connected layer was used to classify the features and obtain a category label for each pixel to achieve the semantic segmentation of the gear defects.

### 2.1. Multiscale Feature Reconstruction

As the task of gearbox gear surface defect detection requires the prediction and segmentation of targets of different sizes to enhance the degree of fusion of the target space and semantic information in the respective regions and improve the ability to interact with information between different windows, in this study, the Swin Transformer was used as a feature extraction module to reconstruct image features and deepen the overall network's understanding of the image information, as shown in Figure 2.

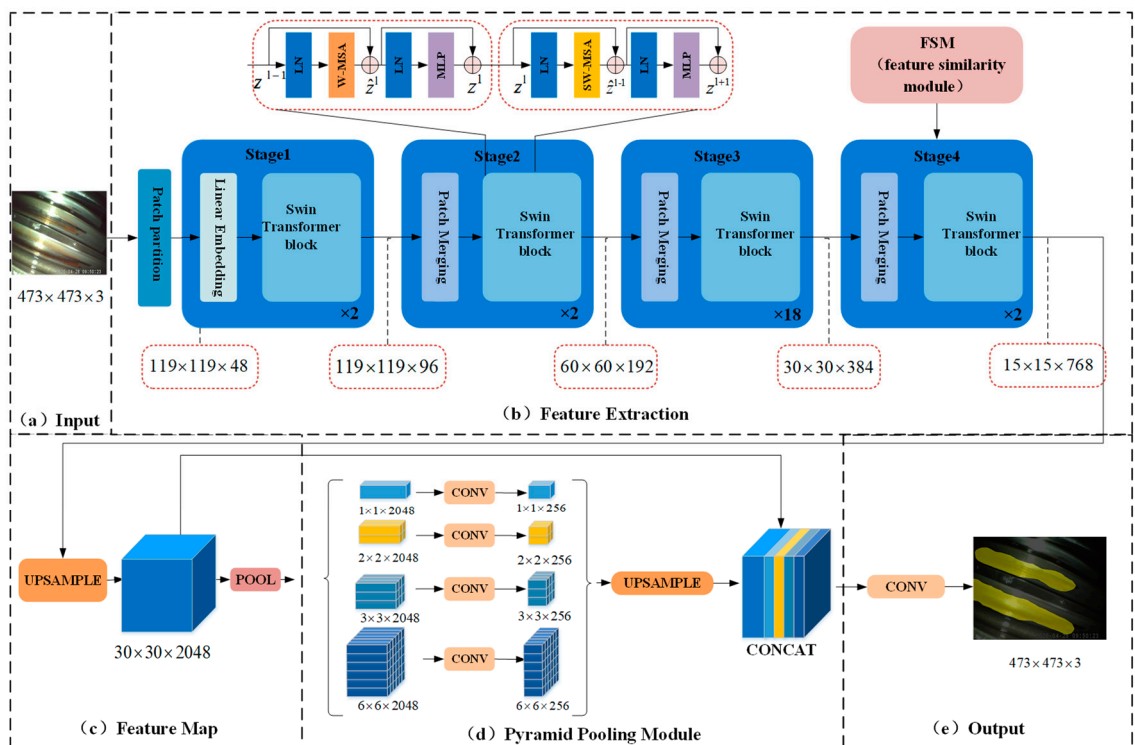

**Figure 1.** Overall framework of the model.

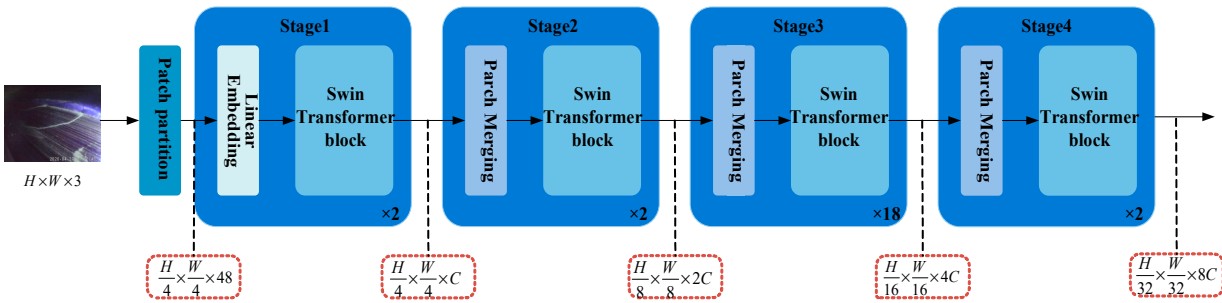

**Figure 2.** Structure of the Swin Transformer network.

The multiscale feature reconstruction comprises multiscale feature extraction and multiscale feature stitching.

(1)    Multiscale feature extraction

The core of multiscale feature extraction is the Swin Transformer block, as shown in Figure 3, whose backbone consists of a multilayer perceptron, Window Multi-head Self-Attention (W-MSA), Shifted Window-based Multi-head Self-Attention (SW-MSA), and Layer Normalization (LN).

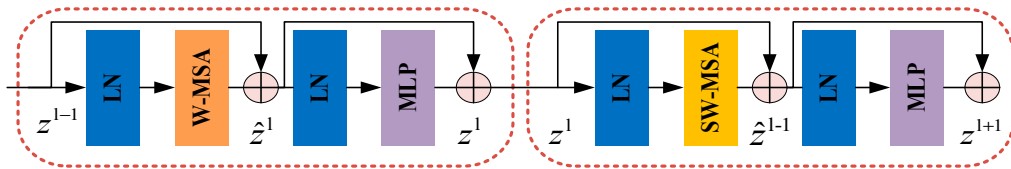

**Figure 3.** Swin Transformer block.

The output of each node is represented as

$$\hat{Z}^l = \text{W\_MSA}\left(\text{LN}\left(Z^{l-1}\right)\right) + Z^{l-1} \tag{1}$$

$$Z^l = \text{MLP}\left(\text{LN}\left(\hat{Z}^l\right)\right) + \hat{Z}^l \tag{2}$$

$$\hat{Z}^{l+1} = \text{SW\_MSA}\left(\text{LN}\left(Z^l\right)\right) + Z^l \tag{3}$$

$$Z^l = \text{MLP}\left(\text{LN}\left(\hat{Z}^{l+1}\right)\right) + \hat{Z}^{l+1} \tag{4}$$

$\hat{Z}^l$ and $Z^l$ denote the feature outputs of SW-MSA and MLP in the $l$th module, and $Z^{l-1}$ denotes the output features of the $l-1$ layer.

The W-MSA module can effectively reduce the computational effort by dividing the feature map into windows of size M × M (M = 4 in Figure 4) and then performing self-attentive computations individually for each window. The SW-MSA module overcomes the difficulty of extracting high-level semantic information by linking adjacent but non-overlapping windows in the upper layer, thereby increasing the perceptual field and capturing high-level semantic information of the image. Therefore, the W-MSA and SW-MSA modules must be used alternately to enable adjacent windows to transfer information and solve the problem of missing features. As shown in Figure 4a, by using the shifted-window method, the four windows of the W-MSA can be transformed into the nine windows of the SW-MSA. During the operation, self-attention calculations must be performed inside each window, which places high demands on the operation procedure; therefore, an efficient batch calculation with a shift configuration was used [25]. This involved a cyclic shift to the upper left. The window was shifted so that each pixel was only computed with the pixels in the region it was currently in. Finally, the data were shifted back to their original position, as shown in Figure 4b.

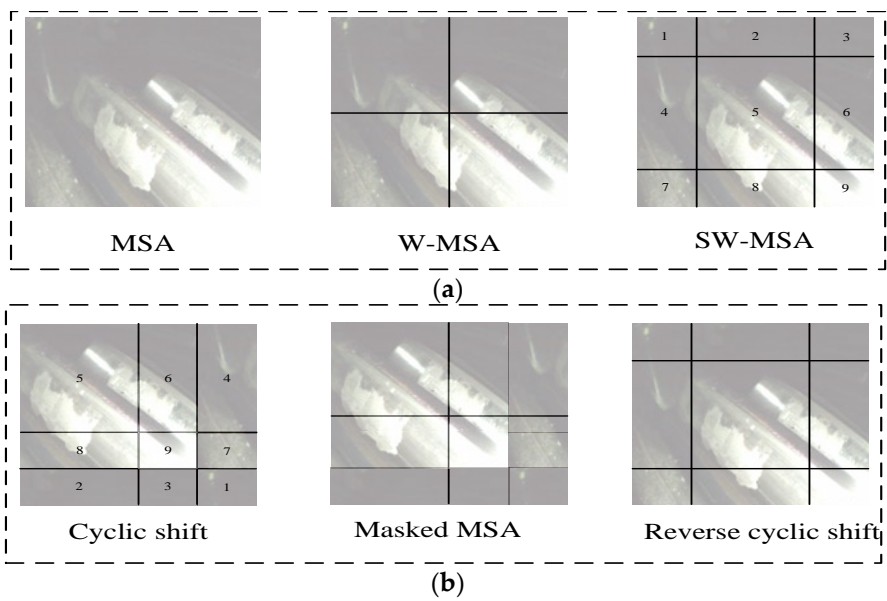

**Figure 4.** Description of the shift window process. (**a**) Shifted Windows Multi-head Self-Attention. (**b**) Shift configuration batch calculations.

(2)    Multiscale feature

Multiscale feature stitching uses the feature map formed by feature extraction to form a feature map containing multiscale information, using Patch Merging to complete multiscale feature reconstruction, as shown in Figure 5. First, the Swin Transformer block was used as the input to the feature map. Then, the blocks of pixels at the same relative position were

segmented and connected along the depth direction, and the obtained results were input to the normalization layer for normalization.

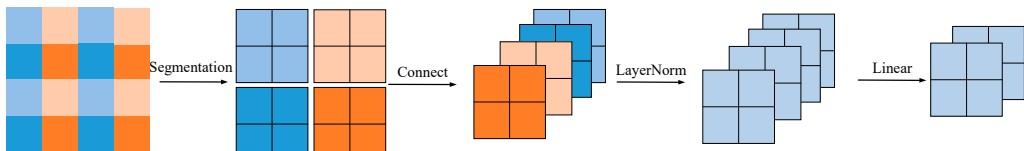

**Figure 5.** Schematic diagram of multiscale feature stitching.

The Swin Transformer has three models: Swin-T, Swin-S, and Swin-B, which have different numbers of output channels (C) and of superimpositions of the cascade module 3. The model parameters are presented in Table 3.

**Table 3.** Parameters of the Swin Transformer model.

| Models | Image Size | C | Stage 1 | Stage 2 | Stage 3 | Stage 4 |
|--------|-----------|-----|---------|---------|---------|---------|
| Swin-T | 473 × 473 | 96 | 2 | 2 | 6 | 2 |
| Swin-S | 473 × 473 | 96 | 2 | 2 | 18 | 2 |
| Swin-B | 473 × 473 | 128 | 2 | 2 | 18 | 2 |

### 2.2. Feature Selection Module

Owing to the small sample size of certain defect data, it is easy to obtain similar and repeated feature information, which does not allow the detection of the differences between defects, interfering with the extraction of defect details by the feature extraction network. Therefore, to accurately assign the required feature details and simultaneously exclude redundant feature maps, the multiple attention mechanism of Stage 4 (the last module) of the Swin Transformer was fully utilized to screen more representative feature maps and build the Feature Selection Module (FSM).

Specifically, a single-headed attention map was selected as $A_i \in R^{N \times N}$, where $N$ represents the length of the input sequence, and each column of this attention map matrix is averaged to obtain the average attention map $A_{avg}$ (length $N$), with the length of the sequence unchanged. The elements in the average attention represent the response of the corresponding image block to the feature extraction model; a larger weight corresponds to a higher degree of utilization of that part of the feature. The feature region with the largest weight value was selected according to $A_{avg}$ and used as a candidate feature image block. The specific formula is given by Equations (5) and (6), where $A_{(i,j)}$ denotes the attention weight of the $i$th row and $j$th column in the single-headed attention matrix $A$, and $\rho_i$ denotes the importance score of the $j$th feature to the model.

$$\rho_j = \frac{1}{N} \sum_{i=1}^{N} A_{(i,j)} \tag{5}$$

$$A_{avg} = [\rho_0, \rho_2, \rho_3 \ldots \rho_N] \tag{6}$$

Because the blocks of the Swin Transformer are all multi-head attention blocks, each single-headed attention mechanism of a block in Stage 4 was averaged to obtain a block of feature candidates; the first block had a total of $X$ single-headed attention mechanisms, resulting in a total of $X$ candidate blocks. In Stage 4, there were two blocks with the same number of attention heads; thus, $2X$ blocks could be obtained. Finally, the $2X$ blocks were pooled and averaged to obtain a global representation $\hat{X}$, which was the input of the contrastive loss function.

The purpose of contrastive loss is to increase the inter-class disparity and reduce the intra-class variation of feature blocks to capture a more discriminative feature map [26]. The contrastive loss function is defined as follows:

$$\text{L}_{CON} = \frac{1}{2N} \sum_{n=1}^{N} \left( Y D_W^2 + (1 - Y)\max(m - D_W, 0)^2 \right) \tag{7}$$

$$Y = \begin{cases} 0 & \hat{X}_1 \neq \hat{X}_2 \\ 1 & \hat{X}_1 = \hat{X}_2 \end{cases} \tag{8}$$

where $Y$ is the discriminant variable that measures the similarity of the sample blocks, $\hat{X}_1$ and $\hat{X}_2$ denote the feature pairs ($Y = 1$ when the feature pairs are similar), $N$ represents the batch size, $D_W$ represents the Euclidean distance between two samples, and $m$ represents the set threshold.

$$D_W\left(\hat{X}_1, \hat{X}_2\right) = \hat{X}_1 - \hat{X}_2 = \left( \sum_{i=1}^{p} \left(\hat{X}_1^i - \hat{X}_2^i\right)^2 \right)^{\frac{1}{2}} \tag{9}$$

When $Y = 1$, the loss function can be obtained from Equation (7) as

$$\text{L}_S = \frac{1}{2N} \sum_{n=1}^{N} Y D_W^2 \tag{10}$$

Accordingly, when two features are similar, the loss function is optimized to reduce the distance between similar features.

When $Y = 0$, the loss function can be obtained from Equation (7) as

$$\text{L}_D = \frac{1}{2N} \sum_{n=1}^{N} \max((m - D_W, 0))^2 \tag{11}$$

Accordingly, when the difference between two features is large, the loss function is optimized to increase its value, and the model parameters are adjusted via backpropagation to increase the difference between classes and reduce the difference within classes for enhancing the discriminative ability of the network.

### 3. Experiments and Analysis of the Results

*3.1. Experimental Procedure*

PyTorch—a mainstream open-source framework for deep learning—was used in the experiment. The experimental environment was Windows 10 with an E5-2640 2.40-GHz CPU and an NVIDIA Quadro P2200 GPU. The model training steps were as follows:

(1) The 1141 dentate images in the dataset were divided into a training set, a validation set, and a test set at a ratio of approximately 7:2:1.

(2) A multiscale feature reconstruction model was developed, pretraining weights were applied, and the initial learning rate was set as 0.001. The learning rate was automatically adjusted by the step size. The ReLU activation function was used, and the number of epochs was 300.

(3) The gear image validation samples were input, and the detection results of the segmentation model were evaluated every seven epochs using the loss function evaluation metrics.

(4) The parameters were adjusted according to the evaluation results to obtain the optimal network model.

In step (2), the learning rate was updated automatically at 15 iterations of the network so that a higher learning rate could be used in the initial training phase to quickly find the direction of the gradient descent, after which the learning rate was automatically updated to obtain the optimal parameter value of the network. In step (3), the model detection

results were evaluated every seven epochs to effectively monitor the network training and facilitate the model review and breakpoint retraining. To evaluate the performance of the three models U-Net, PSPNet (ResNet50), and DeepLabv3+, these models were trained sequentially according to steps (3) to (4), and their detection results were compared with those of the proposed multiscale feature reconstruction model.

*3.2. Wind Turbine Gearbox Surface Defect Dataset Production*

The study of wind turbine gearbox gear surface defect detection relies on a large number of images of gear surface defects, but there is no publicly available dataset containing images of similar gear surface defects; so, the most important aspect of this study was to build a high-quality gear surface image dataset. In this study, a GE industrial endoscope was used to obtain real defect images from a wind turbine gearbox, and the original images were extended by geometric and color space transformations in order to improve the accuracy of the model. The specific methods included horizontal flip, histogram equalization, noise addition, and random rotation. After image enhancement, there were 1350 images, which were classified into five types of defects: rust, deviation, bonding, spalling, and cracking. Finally, these images were filtered to obtain 1141 valid images. The particular defect types are shown in Figure 6.

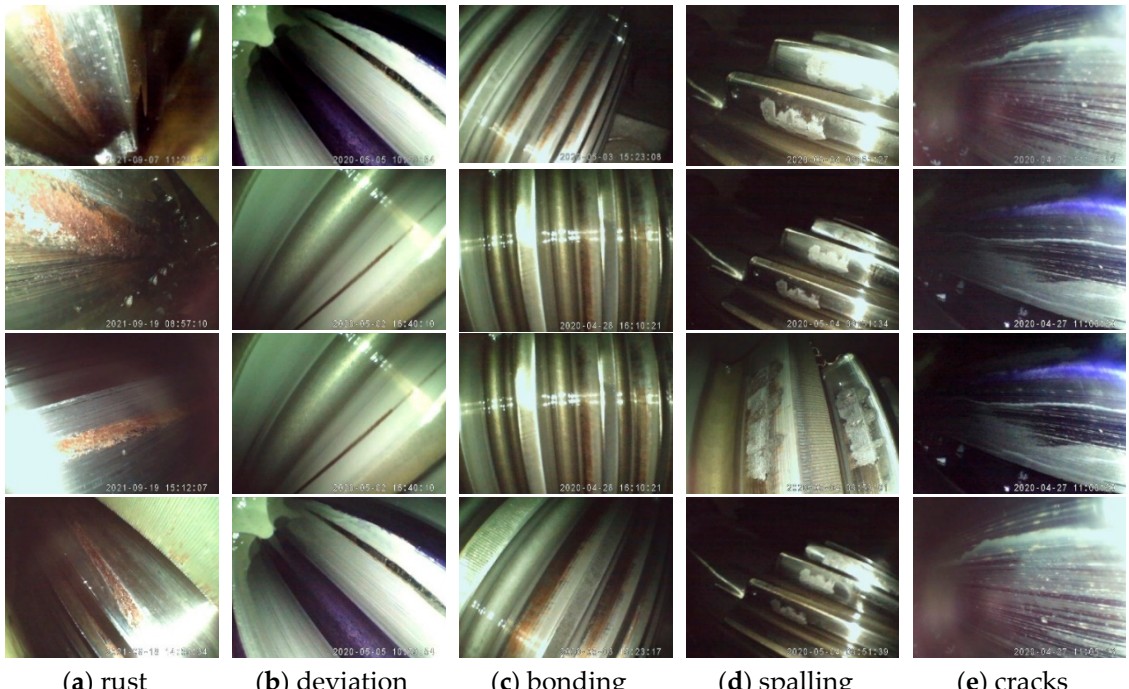

(**a**) rust     (**b**) deviation     (**c**) bonding     (**d**) spalling     (**e**) cracks

**Figure 6.** Example of defects' images.

A total of 1144 valid images were labeled, each with a single defect according to the classification. The open-source tool LabelMe was used for labeling, and the files generated after labeling were converted into the standard format of the VOC dataset used by the network. The dataset was randomly divided into training, validation, and test sets at a ratio of approximately 7:2:1 before training; the numbers of images in these three sets were 800, 228, and 116, respectively. The specific annotations are presented in Table 4.

**Table 4.** Statistical table of annotation information of the dataset.

| Labeling Category | Total | Number of Labels | Rust | Deviation | Bonding | Cracks | Spalling |
|---|---|---|---|---|---|---|---|
| Quantity (sheets) | 1141 | 1141 | 250 | 239 | 251 | 137 | 264 |
| Percentage of | - | 100% | 21.98% | 20.97% | 22.00% | 12.05% | 23.00% |

### 3.3. Defect Detection Effect Evaluation Index

In image semantic segmentation research, evaluation metrics are commonly used to assess algorithms. The most important metrics are the mean Pixel Accuracy (mPA) and the mean Intersection over Union (mIoU). The mIoU represents the average sum of the pixel accuracies of all categories, and the mPA represents the average sum of the accuracies of each correctly classified category.

The Intersection over Union (IoU) represents the ratio of the intersection of the predicted results of the detection model and the true value for each category in the merged set. It is given by Equation (12), where $P_{ii}$ represents the number of correctly detected pixels. In this study, the correct pixels were the defective pixels, and $\sum_{j=0}^{5} P_{ij} + \sum_{j=0}^{5} P_{ji} - P_{ii}$ denotes the number of all pixels, which was calculated as follows:

$$\text{IoU} = \frac{\sum_{j=0}^{5} P_{ii}}{\sum_{j=0}^{5} P_{ij} + \sum_{j=0}^{5} P_{ji} - P_{ii}} \tag{12}$$

The one-way setting is primarily used for the segmentation of few-sample gear defects, i.e., there is only one target defect in the foreground. In this case, the background occupied a large image area, which easily leads to the dominance of the background pixels, resulting in an inaccurate IoU evaluation metric that does not visually represent the performance of the model segmentation. For this reason, mIoU was proposed as an evaluation index; it is more reasonable as a gear defect evaluation index because its calculation does not include the background class but considers the average of all classes of IoU.

In this study, to describe the segmentation performance of the model intuitively, the mIoU was selected as the evaluation index, and the number of categories was six (five categories of defects + one category of background). The mIoU is as follows:

$$\text{mIoU} = \frac{1}{5+1} \sum_{i=0}^{5} \frac{P_{ii}}{\sum_{j=0}^{5} P_{ij} + \sum_{j=0}^{5} P_{ji} - P_{ii}} \tag{13}$$

The pixel accuracy (PA), which represents the proportion of pixels that are correctly detected, is obtained using Equation (14), where $P_{ii}$ represents the number of correctly detected pixels, and $\sum_{i=0}^{5} \sum_{j=0}^{5} P_{ij}$ is the total number of pixels.

$$\text{PA} = \frac{\sum_{i=0}^{5} P_{ii}}{\sum_{i=0}^{5} \sum_{j=1}^{5} P_{ij}} \tag{14}$$

In gear defects, fewer defect pixels have more background pixels, which has less impact on the results; therefore, the PA is not applicable to the detection of gear defects. For this reason, the accuracy of the model detection can be effectively assessed by using the mPA to calculate the proportion of pixels of each defect that are correctly classified relative to all pixel points of that class, after which the average value is taken; thus, the accuracy of each defect is calculated. The mPA is given by Equation (15).

$$\text{mPA} = \frac{1}{5+1} \sum_{i=0}^{5} \frac{P_{ii}}{\sum_{j=1}^{5} P_{ij}} \tag{15}$$

### 3.4. Comparison of the Results and Analysis

The Swin-T, Swin-S, and Swin-B were used in the algorithm of this study for the gear surface defect segmentation task, and the mIoU and mPA results for the above models are presented in Table 5. As shown, with an increase in the model size, the mIoU and mAP improved; the mIoU and mAP with Swin-B as the backbone were improved by 0.9% and 2.24%, respectively, compared with those with Swin-T.

**Table 5.** Comparison results of different Swin Transformer models.

| Model | Backbone | mIoU (%) | mPA (%) |
|-------|----------|----------|---------|
| | Swin-T | 73.87 | 81.77 |
| Ours | Swin-S | 74.23 | 83.92 |
| | Swin-B | 74.77 | 84.01 |

To investigate the extraction of image features with different magnitudes of Swin-T, Swin-S, and Swin-B, the detection performance of the model was visualized and analyzed using Gradient-weighted Class Activation Mapping (Grad-CAM) [27], as shown in Figure 7. In the segmentation of cracks (rows 1 and 2 both show crack detection), Swin-T showed the highest responsiveness to crack defects when there was a large gap between the crack and the background (row 1). However, when the difference between the crack and the background was not obvious (row 2), the three networks had low responsiveness; that is, the samples were not easily analyzed, and this class of samples had a negative impact on the network convergence. Among the three Swin Transformer models, Swin-B showed the most accurate distribution range of high-response areas and could effectively focus and cover most of the defect areas with the ability to learn the characteristics of the gear surface defects, whereas the high-response areas were not obvious for Swin-S. For Swin-T, the sub-response areas not only covered the defect areas but also had the problem of attentional bias. The above results showed that the detection performance of the Swin-T, Swin-S, and Swin-B models was enhanced sequentially, and that of the Swin-B model was higher than those of the other models in terms of recognition efficiency and generalization.

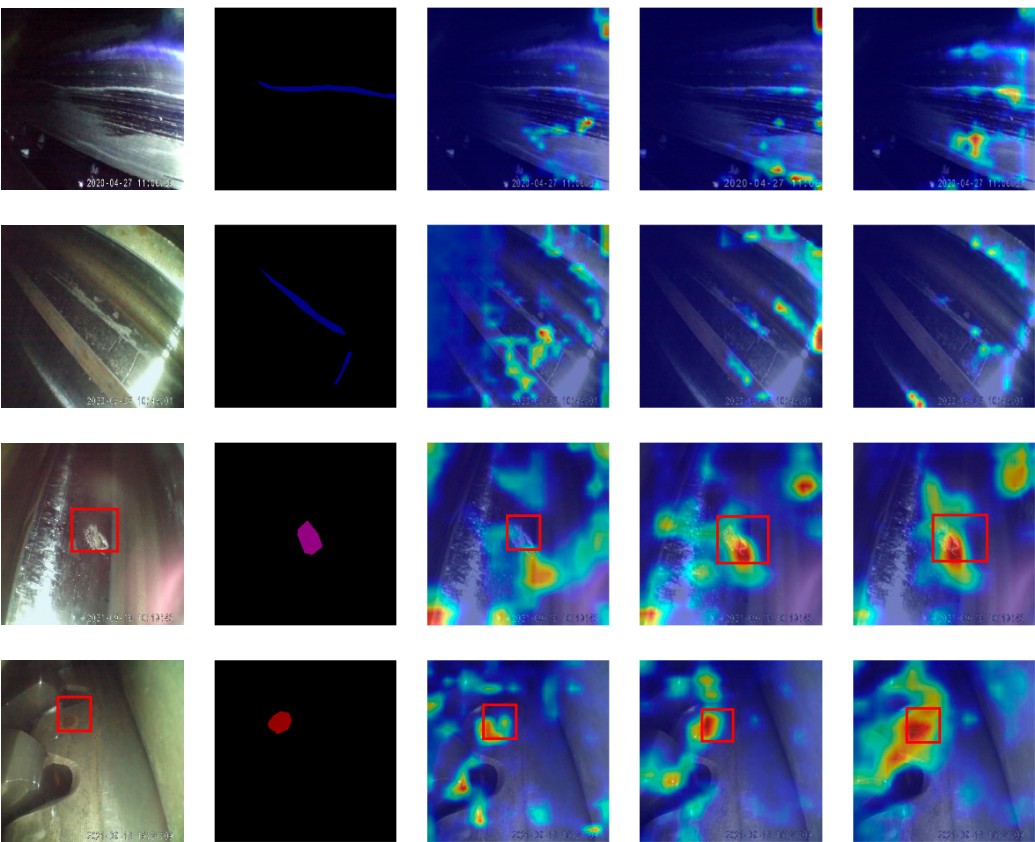

**Figure 7.** Thermal comparison of different Swin Transformer models. Column 1 shows the original image of the gear surface defect (marked by red boxes), column 2 shows the labeled image (marked by differently colored blocks), and columns 3–5 show heat maps of Swin-T, Swin-S, and Swin-B, respectively.

To investigate the effect of adding the FSM on the detection of each defect, the quantitative detection results were examined, as shown in Figure 8. In the spalling defect detection, the accuracy and mIoU were almost unchanged after the addition of the FSM, indicating that the FSM provided no obvious advantage for this type of defect. In the detection of bonding and rust, which are more similar to the background and therefore cause difficulties in detection, the FSM led to an improvement of 3% and 2%, respectively; thus, the FSM had the best performance for these types of defects. In the crack defect detection, owing to the small sample size of the crack defects, the addition of the FSM improved the performance, although a few pixels were missed. In summary, the FSM could filter the image features better and showed a better detection performance for most gear surface defects, but when detecting gear surface defects with a very small number of images such as crack defects, the FSM efficacy was affected by the number of images, and the performance was reduced because there were fewer features available for the model to learn.

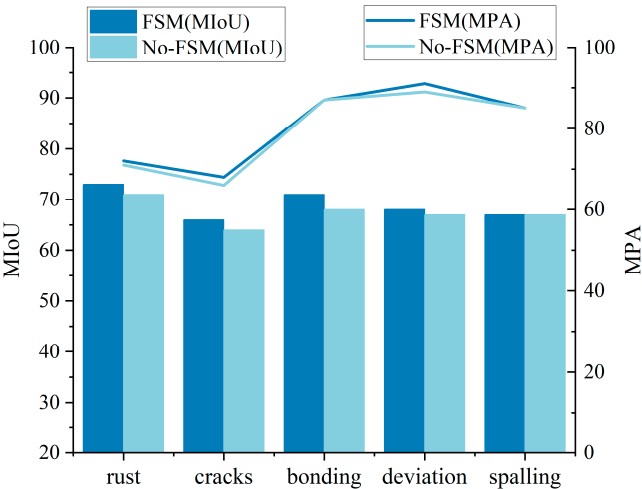

**Figure 8.** Effect of effective feature selection on mIoU and accuracy.

To evaluate the effect of each module on the network detection performance, ablation experiments were designed. The experimental results are presented in Table 6, where"√" indicates that the module is used and "---" indicates that it is not used. The mIoU and mPA of the model were 73.77 and 80.31, respectively, for the baseline algorithm, i.e., without the FSM. For Model 2, with the addition of the FSM, the mIoU and mPA were improved by 0.12% and 3.34%, respectively. For Model 3, with the addition of Swin-B, the mIoU and mPA were improved by 1.00% and 3.70%, respectively. For Model 4, with the addition of both the FSM and Swin-B, the mIoU and mPA were improved by 1.21% and 3.88%, respectively. According to the experimental results, the detection network with a combination of Swin-B and the FSM could achieve a higher detection accuracy than when using Swin-B or the FSM alone.

**Table 6.** Ablation experiments.

| Models | FSM | Swin-B | mIoU (%) | mPA (%) |
| --- | --- | --- | --- | --- |
| Model 1 | --- | --- | 73.77 | 80.31 |
| Model 2 | √ | --- | 73.89 | 83.65 |
| Model 3 | --- | √ | 74.77 | 84.01 |
| Model 4 | √ | √ | 74.98 | 84.19 |

For the established training and validation sets, both the proposed algorithm and the PSPNet were trained for 300 epochs with a batch size of 4. The Adam optimizer was used to optimize the network parameters. The MIoU and loss variation curves of the

proposed algorithm and the PSPNet network for the training set with respect to the number of iterations are shown in Figure 9.

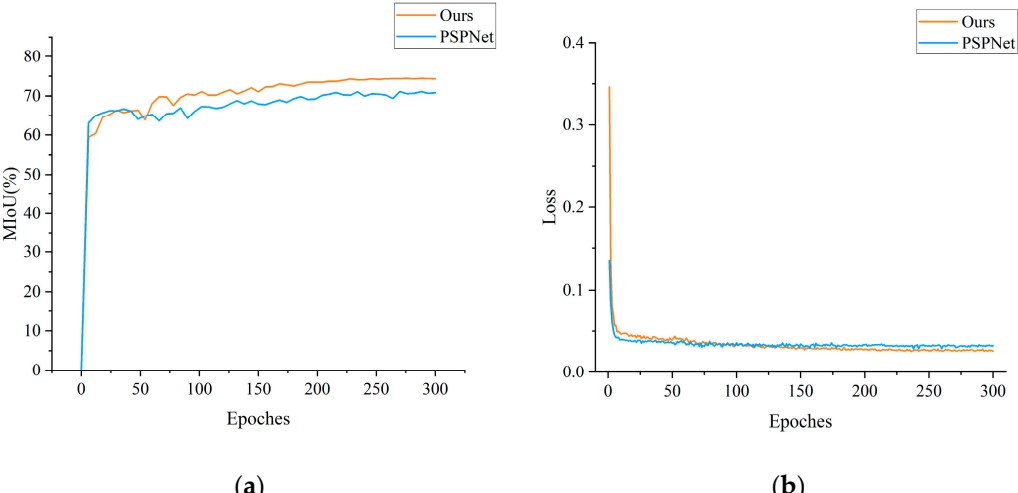

**Figure 9.** MIoU and loss variation. (**a**) Changes in mIoU. (**b**) Variation of the loss value.

As indicated by Figure 9a, the proposed algorithm rapidly increased the model accuracy at the early stage of training; the convergence was faster than that of the PSPNet model, and the accuracy of the validation set segmentation exceeded 70%. At approximately 70 iterations, the mIoU of the proposed algorithm for the validation set fluctuated significantly owing to the influence of the computational resources. The migration learning strategy was used to freeze a large number of parameters of the network before 70 iterations to accelerate the convergence of the network; after 70 iterations, the network parameters changed, causing the mIoU curve of the validation set to oscillate, while the loss value also broke the bottleneck and decreased again. The change in loss in Figure 9b indicates that after 270 iterations, the loss values of the two models tended to stabilize; the loss value of the proposed algorithm remained below 0.3, and that of PSPNet remained below 0.6. The statistical analysis of the loss comparison graph and the average cross-merge ratio showed that the detection performance of this model was better than that of the PSPNet model overall and is thus suitable for the gear surface defect detection task.

To evaluate the segmentation efficiencies of the different models, an image test was performed using the test set, and 50 images were selected for each test, including rust, deviation, bonding, cracks, and spalling images. The test results are presented in Table 7. Compared with the original PSPNet network, the proposed model has the advantage of segmenting gear defects in complex backgrounds. It outperformed the PSPNet (ResNet50) by 1.21% and 3.88% for the mIoU and mAP, respectively. By comparing the number of parameters and the inference time on GPU for different models, it was found that the model used in this paper obtained a higher accuracy by sacrificing a smaller number of model parameters and inference time.

**Table 7.** Algorithm comparison results.

| Models | Backbone | mIoU (%) | mPA (%) | Params (M) | Time (s) |
|---|---|---|---|---|---|
| DeepLabv3+ | Xception | 71.06 | 72.01 | 51.3 | 0.8 |
| Unet | ResNet50 | 71.12 | 72.12 | 43.9 | 0.5 |
| PSPNet | ResNet50 | 73.77 | 80.31 | 46.7 | 1.5 |
| Ours | Swin-B | 74.98 | 84.19 | 52.9 | 1.8 |

The comparison results for the proposed algorithms, U-Net, DeepLabv3+, and PSPNet, are shown in Figure 10. The segmentation results obtained in this study were similar to

those of a real label image, with a good segmentation effect. The edge of the segmentation result was fine, showing that the method is suitable for gear surface defect detection.

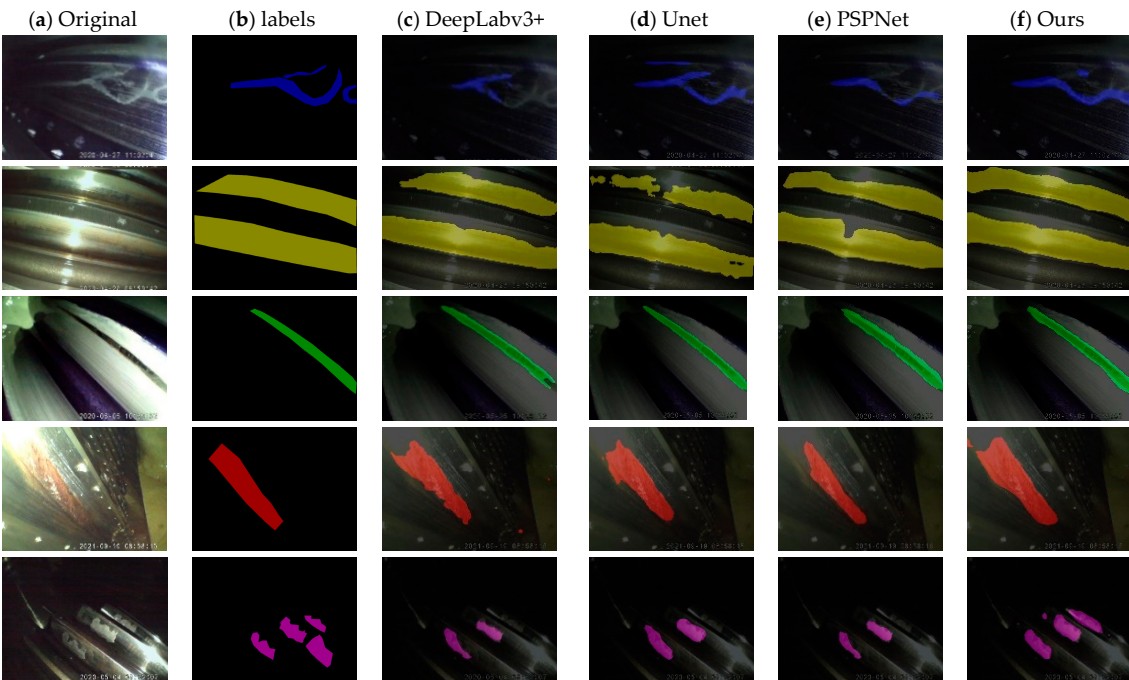

**Figure 10.** Contrast experimental semantic segmentation results.

## 4. Conclusions

This study proposes a gearbox gear surface defect detection method based on a multiscale feature reconstruction model with the robustness and accuracy of an industrialized model for the problem of difficult feature extraction of wind turbine gearbox gear surface image defects. First, we innovatively used Swin Transformer with FSM in the PSPNet network for multiscale feature extraction and feature selection of feature maps, which effectively solved the problem of insufficient feature extraction of gear surface defects. Secondly, we experimentally identified Swin-B as a module for multiscale feature reconstruction, as well as demonstrated that the feature selection module had a better detection performance in the case of relatively similar backgrounds. Finally, we compared the detection results of DeepLabv3+, Unet, PSPNet, and multiscale feature reconstruction and found that the multiscale feature reconstruction detection model showed a higher detection accuracy and can be used for the detection of gear surface defects.

The network model designed in this paper has high detection accuracy but needs to be improved in terms of the number of parameters and inference time. In future work, we will consider lightweighting the model and design a more compact and lighter network structure. In addition, since the fan gearbox gear surface is similar to most metal gear surfaces in shape, this model has some significance as a reference for metal gear surface defect detection in other contexts.

**Author Contributions:** Conceptualization, X.C.; methodology, R.G. and J.D.; software, J.C. and D.L.; formal analysis, H.X.; writing-review R.G. and J.C. All authors have read and agreed to the published version of the manuscript.

**Funding:** This work was supported by the Natural Science Basic Research Program of Shaanxi Province, China [2021JLM-04].

**Data Availability Statement:** The gearbox gear surface dataset in this paper is a homemade dataset and is not disclosed due to its use in subsequent studies.

**Conflicts of Interest:** The authors declare no conflict of interest.

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
