# Peer review of "Wind Turbine Gearbox Gear Surface Defect Detection Based on Multiscale Feature Reconstruction"

_electronics, doi:10.3390/electronics12143039_

Round 1
Reviewer 1 Report
In this work is proposes a multiscale feature reconstruction-based detection method for detecting wind-turbine gearbox surface defects.
The proposal is interesting but some issues must be addressed.
1. The literature review includes some of the most important works that have been published in the field, the most common signals used for assessment are vibration, stator currents and sound; however, some works have reported the use of stray flux signal for performing the fault detection in gearboxes. Can the authors include a brief discussion about that, please consider the following paper, 10.1109/TIA.2022.3174049 .
2. The description of section 3.2 is very restrictive and more details are required for a better understanding.
3. The discussion of the obtained results must be improved in order to highlight the advantages of the proposed method
4. The conclusion section must be rewritten in order the provide the main and most important aspects of this proposal, as well, include future work.
5. What are the advantages and disadvantages in front of other methods, include a brief discussion about that.
Author Response
Thank you for your letter and for the reviewers’ comments concerning our manuscript entitled “Wind-turbine gearbox gear surface defect detection based on multiscale feature reconstruction”. Those comments are all valuable and very helpful for revising and improving our paper, as well as the important guiding significance to our researches.
Please see the attachment for detailed revised responses.

Reviewer 2 Report
This paper proposes a multiscale feature reconstruction-based detection method for wind-turbine gearbox surface defects. The technique uses the Swin Transformer as a backbone network, a Feature Similarity Module to filter important feature sub-blocks, and a fusion of contextual information using the Pyr-amid pooling module to enhance the extraction of tooth surface defect features at different scales. Experimental results show that the improved algorithm outperforms the original PSPNet algorithm by 1.21% and 3.88% for mean intersection over union and mean pixel accuracy, respectively. The method significantly outperforms semantic segmentation networks like U-Net and DeepLabv3+. The proposed method is crucial for wind-turbine maintenance and power security.
The manuscript is well-organized and well-written.
The following can be fixed before publication.
1- What about the running time (execution time) of the method? And it's better to compare the proposed method with other Models.
2- Providing a table that summarizes the related work would increase the understandability of the difference from the previous studies in the "Related Works" section.
3- Some abbreviations are used in the text without giving their expansion.
4- Please add one paragraph about this work's novelty. And please explain it clearly.
5. Add one section (i.e., discussion) about the limitations of this method? How do you overcome them in the future? And Can you use the method for other applications?
Other minor things:
- Highlight the best value in the table
- Reference style
- Conclusions (write at least one paragraph)
- space before and after figures/equations
Author Response

(The authors gave the same response as above.)

Round 2
Reviewer 2 Report
Most of my remarks have been added. In conclusion, it remains inadequate. Please provide a conclusion that is in the form of a narrative and is more thorough than a list of bullet points with no good transition between sentences.
Author Response

(The authors gave the same response as above.)
